# Quantum enhanced feedback cooling of a mechanical oscillator using nonclassical light

Clemens Schäfermeier[1], Hugo Kerdoncuff[1], Ulrich B. Hoff[1], Hao Fu[1], Alexander Huck[1], Jan Bilek[1], Glen I. Harris[2], Warwick P. Bowen[2], Tobias Gehring[1] & Ulrik L. Andersen[1]

Laser cooling is a fundamental technique used in primary atomic frequency standards, quantum computers, quantum condensed matter physics and tests of fundamental physics, among other areas. It has been known since the early 1990s that laser cooling can, in principle, be improved by using squeezed light as an electromagnetic reservoir; while quantum feedback control using a squeezed light probe is also predicted to allow improved cooling. Here we show the implementation of quantum feedback control of a micro-mechanical oscillator using squeezed probe light. This allows quantum-enhanced feedback cooling with a measurement rate greater than it is possible with classical light, and a consequent reduction in the final oscillator temperature. Our results have significance for future applications in areas ranging from quantum information networks, to quantum-enhanced force and displacement measurements and fundamental tests of macroscopic quantum mechanics.

[1] Department of Physics, Technical University of Denmark, Fysikvej 309, 2800 Kgs Lyngby, Denmark. [2] Australian Centre of Excellence for Engineered Quantum Systems, University of Queensland, St Lucia, Queensland 4072, Australia. Correspondence and requests for materials should be addressed to U.L.A. (email: ulrik.andersen@fysik.dtu.dk).

Real-time quantum feedback control[1] often involves time-continuous measurements of a quantum state followed by control of a physical system with the aim of steering that system into a desired state. Such control has been demonstrated in numerous experiments, including the preparation of microwave Fock states[2] and the generation of deterministic entanglement in quantum superconducting circuits[3]. Quantum feedback control also allows the control of motional states via continuous measurements of a probe field followed by actuation back on the motional degree of freedom. It has been used to steer single trapped atoms[4], levitated microspheres[5], micromechanical oscillators[6,7] and suspended kilogram-scale mirrors[8] towards their motional ground states. Furthermore, quantum feedback control was proposed as a means to prepare both mechanical Fock[9] and superposition states[10]. However, to-date, all such experiments have used a classical probe field.

The use of a nonclassical probe field for quantum control can both increase the effectiveness of steering to the desired quantum state and introduce new functionalities. For instance, it can be used to prepare a mechanical oscillator in a non-Gaussian quantum state even in the linear coupling regime[11–13], thereby circumventing the usual requirement to reach the single-photon strong coupling regime[14]. Moreover, quantum feedback control using squeezed light can enable an efficient interface for quantum information networks[15,16], and allow the standard quantum limit of force measurements to be surpassed via squeezing-enhanced measurements[17]. Here we demonstrate the first critical step towards such functionalities by realising a quantum-enhanced cooling rate using a squeezed probe field and real-time feedback control of a mechanical mode.

The basic idea is to continuously monitor the position of the mechanical oscillator and subsequently use the measurement record to steer the oscillator towards the ground state with real-time feedback[18]. The cooling rate is limited by the precision with which the position of the oscillator can be monitored, which in turn is often limited by the noise of the tracking laser beam[19]. Therefore, by engineering the quantum state of the laser beam to reduce the measurement noise, an enhanced cooling rate can be expected. Using squeezed light with variance $V_{sqz}$ to continuously monitor the mechanical motion, a feedback loop with signal detection efficiency $\eta$, an overall detection efficiency $\eta_d$ for the squeezed light, and feedback gain $G_{Fb}$, can lower the temperature to

$$T_{Fb} = \left(1 + \frac{V_d}{8\eta n_{th}C}G_{Fb}^2\right)\frac{1}{1+G_{Fb}}T_0, \qquad (1)$$

where $V_d$ is the detected variance $\eta_d V_{sqz} + (1 - \eta_d)/2$, $T_0$ is the equilibrium temperature of the mechanical oscillator before feedback cooling, $n_{th}$ the corresponding mean phonon occupancy and we have neglected radiation pressure induced heating due to the measurement process. Considering a cavity optomechanical system with cavity energy decay rate $\kappa$ and mechanical dissipation rate $\Gamma_m$, the linearized interaction is characterized by the cooperativity $C = 4g_0^2 N_c/\kappa\Gamma_m$, where $g_0$ is the single-photon optomechanical coupling strength and $N_c$ the mean intra-cavity photon number. As the feedback gain is increased, the temperature drops, reaching a minimum for $G_{Fb}^{opt} = \sqrt{1 + 8\eta n_{th}C/V_d} - 1$. At higher gain, the oscillator begins to heat up due to the imparted measurement noise. In essence, the application of squeezed measurement noise acts to shift this onset of heating to higher gain values, thereby allowing cooling to temperatures below the shot noise imposed limit, as illustrated in Fig. 1d. Alternatively, the improved cooling efficiency can be appreciated as a consequence of an increased measurement rate $\mu = C \cdot \Gamma_m/2V_d = 2N_c g_0^2/\kappa V_d$, resulting from the squeezing-enhanced signal-to-noise ratio of the transduced mechanical

motion[20]. In comparison to a coherent state probe, the improvement is directly given by the degree of squeezing as $\mu_{sqz}/\mu_{coh} = (2V_d)^{-1}$. As the measurement rate is intimately related to the ability to achieve ground state cooling, requiring $\mu > \Gamma_{th} \sim \Gamma_m n_{th}$, squeezing-enhanced feedback cooling can greatly relax the structural requirements for the mechanical oscillator and thereby facilitate the preparation of macroscopic oscillators in their motional ground state. It should be noted that improved cooling can be also realised by increasing the power of the probe beam. However, it is challenging, in practice, to achieve cooling rates sufficient for ground state cooling due to localized bulk heating introduced by absorption of the probe field.

In this proof-of-principle experiment, we probed a microtoroidal resonator with $V_d = 1.9$ dB vacuum noise suppression. Starting at room temperature, a selected mechanical mode was cooled to 130 K. Compared with the usage of a coherent probe, the improvement in cooling was $>12\%$.

## Results

**Implementation of the optomechanical system.** The experimental setup for squeezed light enhanced feedback cooling is shown in Fig. 1a and a scanning electron micrograph of the optomechanical system—a microtoroidal resonator—is displayed in Fig. 1b.

To demonstrate the principle of quantum-enhanced feedback cooling, we chose to work with the fundamental flexural mode of the microtoroid, highlighted in Fig. 1c. The mechanical resonance frequency of the mode was $\Omega_m/2\pi = 6.13$ MHz, well within the 9 MHz bandwidth of our feedback loop, and its effective mass was determined to be 10 µg via finite element modelling. From the optically read out position power spectral density, shown in Fig. 1c the damping rate was characterized to be $\Gamma_m/2\pi = 13$ kHz. We note that the predominantly vertical motion (shown in the finite element model in Fig. 1c) of the fundamental flexural mode reduces the strength with which it couples to the optical field. While the coupling is sufficient for our proof-of-principle demonstration, stronger coupling—sufficient to enter the quantum-coherent coupling regime[21]—can be achieved using a radial breathing mode. This was precluded in our case, since the resonance frequency $>50$ MHz of this mode laid outside of our available control bandwidth.

**Cooling with a coherent- or squeezed probe.** In the first experiment, we implemented feedback cooling using coherent states of light with a power of 8.5 µW. The results are illustrated in Fig. 2a (left), where we cooled from a temperature of 295 K down to the limit set by the imprecision noise corresponding to the quantum noise of a coherent state. We carefully characterized the noise floor through attenuation measurements and balanced detection to verify that it is of true quantum origin, that is, pure vacuum noise. As the feedback gain is increased, the temperature of the mechanical mode is decreased to 149 K. Increasing the gain further leads to heating of the oscillator as the shot noise of the probing field dominates the control[22].

The demonstration of squeezed light enhanced cooling is presented in Fig. 2a (right). Here, we used the same coherent excitation as for the coherent state cooling experiment, but the quantum fluctuations of the probe beam were now suppressed below vacuum noise. The actual noise suppression of the generated squeezed state was $V_{sqz} = 8$ dB (see Supplementary Methods), but as a result of evanescent coupling and propagation losses in the tapered fibre, the measured noise reduction was limited to $V_d = 1.9$ dB below the vacuum noise. Despite the losses a considerable increase in measurement rate of $\mu_{sqz}/\mu_{coh} = 1.55$ was achieved, and by applying the electronic feedback a clear suppression of the transduced thermally excited

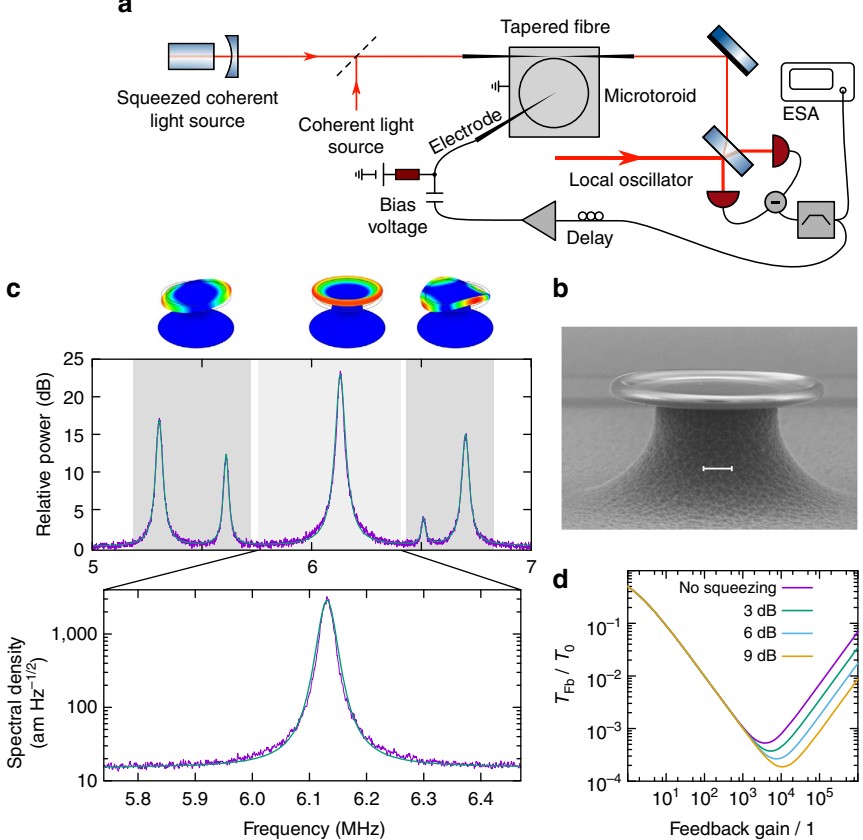

**Figure 1 | Experimental and conceptual details. (a)** Sketch of the experimental setup. The mechanical motion in a microtoroid was sensed using a squeezed or coherent state coupled evanescently to the microtoroid via a tapered fibre. The electrical output of a homodyne detector was bandpass filtered around the mechanical resonance frequency, delayed, amplified and fed back to the mechanical resonator via an electrical actuator. An electrode tip held close above the toroid combined with a grounded base plate formed the actuator. ESA, Electrical Spectrum Analyser. **(b)** Scanning electron microscope picture of the microtoroidal resonator with a radius of 37.5 µm and an optical $Q$ value of $1 \times 10^6$. The scale bar corresponds to 10 µm. **(c)** Mechanical spectrum of the microtoroid measured with a coherent probe beam of 50 µW (equal to an intra-cavity photon number of $N_c = 5.1 \times 10^4$). From the measurement we infer a cooperativity of $C = 5.2 \times 10^{-4}$. The range from 5 to 7 MHz is magnified to highlight the spectrum as it was detected after the bandpass filter. By means of a finite element method analysis, the mechanical modes had been computed. Each mode corresponds to the resonance shown underneath. The first and the third mode are non-degenerate due to asymmetries in the fabrication. **(d)** Example of the achievable temperature improvement in dependence of the degree of squeezing. For the calculation, we assumed $n_{th} = 10^4$, $C = 10^3$, unity efficiencies and a coupling efficiency of 0.1.

mechanical displacement fluctuations was observed, ultimately reaching the squeezed noise level well below shot noise. As a direct result of employing quantum-enhanced feedback cooling, the mechanical mode was cooled to an effective temperature of 130 K, >12% lower than the minimum temperature achieved using coherent light.

A complementary characterization of the demonstrated cooling scheme is provided by examining the time-domain phase space trajectory of the mechanical oscillator. Due to the large bandwidth of the homodyne detector compared with the mechanical dissipation rate, this could be monitored in real-time by simultaneous down-mixing of the homodyne photo-current with two in-quadrature signals and subsequent low-pass filtering[20]. The recorded thermal trajectories are visualized in Fig. 2b. Both probing strategies result in a significant confinement of the oscillator's random excursions in phase space when subject to feedback cooling. While the enhancement by using squeezed light is not directly obvious from the phase space trajectories (Fig. 2b) the effect is more pronounced by considering the marginal quadrature distributions (Fig. 2c), revealing a 12.6% reduction in the variance of the cooled oscillator's position for the squeezed light probe, which is in good agreement with the temperature reduction.

The resulting temperature estimates as function of inferred feedback gain are presented in Fig. 3. A clear cooling improvement is observed for increasing gain until the thermal noise spectrum reaches the measurement imprecision noise level, after which the mechanical oscillator begins to heat up due to measurement noise being imprinted on its motion[22,23]. As expected, the reduced imprecision noise of the squeezed probe shifts the onset of heating towards larger feedback gain values, allowing quantum-enabled cooling to temperatures below the limit set by shot noise.

## Discussion

The absolute cooling achieved in the present demonstration is only modest compared with state-of-the-art[7], the main limitations being the relatively poor optomechanical cooperativity provided by the flexural mode and the requirement for operating in the under-coupled regime to preserve squeezing.

Considerable improvements in cooling performance can be expected by using an optomechanical system with enhanced cooperativity, and realising higher coupling and detection efficiencies for the squeezed mode. For instance, operating the system of Wilson et al.[7] in the under-coupled regime ($C/n_c = 0.62$ at a coupling efficiency of 0.028, where $n_c$ is the intra-cavity

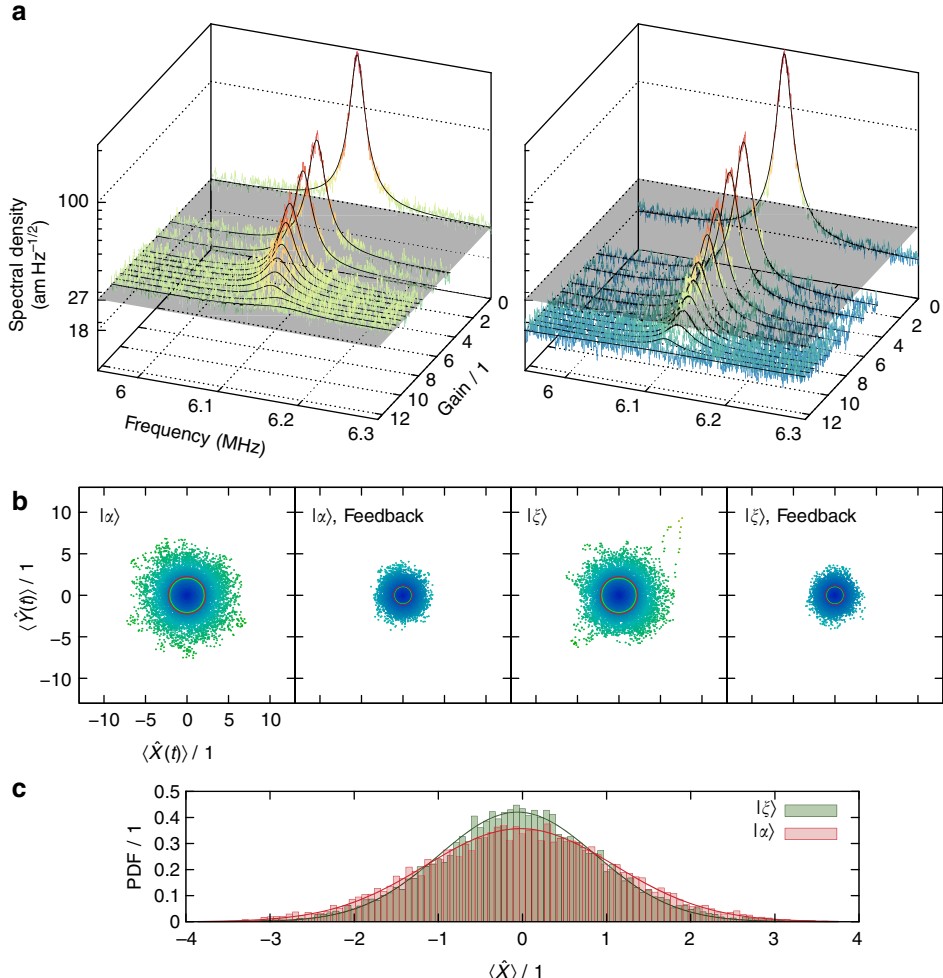

**Figure 2 | Experimental data. (a)** Evolution of the resonance under different gain settings. Left: Coherent state probe. Right: Squeezed state probe. Spectra are corrected for detector dark noise. The grey plane represents the optical shot noise level. The spectra are arranged proportional to the electronic gain that was set for the recording. Solid black lines stem from fits to a Lorentzian distribution. **(b)** Phase space trajectory of the mechanical oscillator's position, normalized to shot noise. As the bandwidth of the detector was much larger than the mechanical dissipation rate, it was possible to monitor the thermal evolution of the oscillator in real-time. $|\alpha\rangle$ and $|\xi\rangle$ refer to the coherent- and squeezed-state probe, respectively. Thin red circles represent the standard deviation of the distribution recorded with $|\alpha\rangle$, green circles visualize the same quantity when the mechanical mode is probed with $|\xi\rangle$. The presented signals are low-pass filtered around the resonance frequency $\Omega_m$ with a bandwidth equal to mechanical dissipation rate $\Gamma_m$. **(c)** The histogram shows the marginal distribution along the $\hat{X}$ quadrature of the cooled mechanical mode, comparing the squeezed (green) and the coherent (red) probe. Solid lines represent fits to the data, assuming a normal distribution, such that the vertical axis is scaled to be a probability density function (PDF), with the same normalization to unity as in the phase space plots.

photon number of $10^4$) so that the injected squeezed light reflects from the cavity with high efficiency, with unity detection efficiency, and starting from 650 mK, enables cooling to an occupation number $n_{th} = 1.7$ phonons. Using a squeezed probe with an input squeezing of 9 dB, the occupation number could be lowered to $n_{th} = 0.4$. We note that achieving close to unity detection efficiency is challenging for current quantum optomechanics experiments. This presents a serious constraint for quantum feedback control experiments. Indeed, with the experimentally realised efficiency reported by Wilson *et al.* of $\eta = 23\%$, even when achieving a cooperativity $C \gg n_{th}$, the minimum mechanical occupancy that could be achieved with coherent light[20] is $n_{min} = 1/(2\sqrt{0.23}) - 1/2 = 0.5$. Interestingly, while squeezed light is degraded by inefficiencies, it also provides a mechanism to mitigate them in feedback control experiments. If amplitude- rather than phase squeezing is used, the contribution of vacuum noise entering the phase quadrature due to inefficiencies is suppressed relative to the amplified noise of the

phase quadrature. This results in a higher effective efficiency for feedback control experiments of $\eta_{eff} = (1 + (1 - \eta)/(2V_{sqz}\eta)^{-1})$, where we assume a minimum uncertainty state with squeezing variance $V_{sqz}$ (Supplementary Note 2). We see that when $V_{sqz} \gg (1 - \eta)/(2\eta)$, $\eta_{eff}$ approaches unity. As an example, assuming an intra-cavity phase anti-squeezing variance 9 dB above shot noise, the effective feedback cooling efficiency of Wilson *et al.* could be increased to $\eta_{eff} = 0.7$, allowing, in principle, feedback cooling to an occupancy of $n_{min} = 0.1$.

We anticipate that the full benefit of squeezing-enhanced feedback cooling could be harnessed using recently developed tethered membrane mechanical oscillators[24] in conjunction with a high-quality optical cavity, or alternatively, by implementation in state-of-the-art optomechanical systems in the microwave regime[25].

Our demonstration of enhanced real-time quantum feedback exploiting squeezed-state enables a number of improvements in the field of quantum optomechanics. Besides the demonstrated

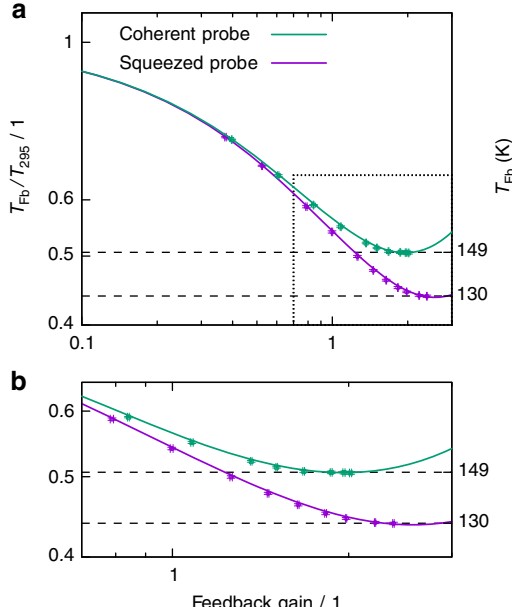

**Figure 3 | Squeezed light enhanced feedback cooling.** (**a**) Temperature ratio between the cooled and uncooled mechanical resonances (left axis) and the absolute temperatures $T_{Fb}$ (right axis) versus the feedback gain. Without feedback cooling, the temperature was 295 K. Points represent measured data. The temperature was determined, in accordance to the equipartition theorem, by integrating the spectral density of the oscillator's displacement. Error bars, calculated by means of a standard uncertainty propagation method, indicate the absolute error (see Supplementary Methods). Solid lines are given by equation (1). (**b**) A zoom into the region framed by a dashed rectangle in the upper plot. For higher gain settings, a squashing effect was observed (see Supplementary Figure 9). This is an effect of the in-loop measurement, correlating the mechanical motion and the measurement noise[23].

cooling effect, it lays the foundation for more advanced schemes including squeezing-enhanced quantum back-action evasion which in turn can be used to prepare mechanically squeezed states in the weak coupling regime and for generation of non-Gaussian mechanical quantum states. As an example, combining 10 dB of squeezing with active feedback in a pulsed backaction evading scheme[13], it is possible to squeeze the mechanical oscillator by 10 dB for an interaction strength of $\chi = 4g_0 \sqrt{N_c}/\kappa = 1$, while for a coherent state input, $\chi \geq \sqrt{10}$ is required to squeeze the mechanics by the same amount. Moreover, supplementing the scheme with photon-subtraction, the mechanics can be driven into a quantum non-Gaussian state even for a modest interaction strength. The control of nonclassical mechanical states will also allow for tests of quantum decoherence models and lead to new sensing technologies such as quantum magnetometry[26], inertial sensing[27] and force microscopy[28].

## Methods

**Experimental details.** Bright squeezed light at 1,064 nm for tracking the mechanical motion was provided by the nonlinear process of cavity-enhanced parametric down-conversion and injected into the toroidal optical cavity with decay rate $\kappa/2\pi = 94$ MHz via evanescent coupling using a tapered single-mode fibre. To allow for evanescent coupling to the microtoroid's optical modes, the fibre was tapered down adiabatically to guide a single-mode 1,064 nm beam. The coupling efficiency to the microtoroid was controlled by changing the relative position between the tapered fibre and the microtoroid, the polarization of the incoupled beam and the temperature of the toroid. The total on-resonance transmission through the optical fibre and toroid was measured to be 50%, mainly limited by scattering losses at the tapered region of the fibre. After interaction with the mechanical mode of the toroid, the phase quadrature of the probe beam was measured by a high efficiency homodyne detector (98%), and the output

photocurrent was split to provide signals for both characterization of the mechanical motion and feedback.

**Feedback loop characteristics.** The feedback loop, consisting of bandpass filtering, delay, and amplification, was designed to exert a viscous damping force on a selected mechanical mode. The required velocity dependence of the applied force was controlled by a tunable signal delay. To actuate the mechanical motion of the toroid, an electric field was generated between a grounded aluminium plate supporting the toroid and a sharp electrode with a tip diameter of about 3 μm (ref. 23). A bias voltage of 295 V was applied to the electrode to polarize the microtoroid, in order to enhance the electrical transduction.

**Temperature estimation.** Following the standard approach[29], the measured and calibrated Lorentzian spectra in Fig. 2a were fitted to a model including the noise squashing effect of in-loop measurements (Supplementary Note 1). For each measurement, the mechanical resonance frequency, feedback gain and measurement noise level were used as free parameters. Substituting the fitted values into equation (1), the effective mode temperature was calculated.

**Data availability.** The data that support the findings of this study are available from the corresponding author on reasonable request.

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

## Acknowledgements

The work was supported by the Lundbeck Foundation (no. R69-A8249), Villum Fonden (no. 1330), the Danish Council for Independent Research (Sapere Aude 4184-00338B and 0602-01686B), the Australian Research Council Centre of Excellence CE110001013 and by the Air Force Office of Scientific Research and the Asian Office of Aerospace Research and Development. W.P.B. is supported by the Australian Research Council Future Fellowship FT140100650. Microtoroid fabrication was undertaken within the Queensland Node of the Australian Nanofabrication Facility.

## Author contributions

U.L.A. conceived the idea while C.S., H.K., T.G. and U.L.A. devised the experiment. C.S. performed the experiment with help from H.K., H.F., U.B.H., G.I.H., T.G. and J.B. C.S. and U.B.H. analysed the data with input from W.P.B. and U.L.A. C.S., U.B.H., T.G., W.P.B. and U.L.A. wrote the paper. C.S. and U.B.H. wrote Supplementary Material with input from H.K. T.G., A.H. and U.L.A. supervised the work.

## Additional information

**Competing financial interests:** The authors declare no competing financial interests.

