## [Peer Review File · Nature Communications]

Editorial Note: *This manuscript has been previously reviewed at another journal that is not operating a transparent peer review scheme. This document only contains reviewer comments and rebuttal letters for versions considered at Nature Communications.*

REVIEWERS' COMMENTS:

Reviewer #2 (Remarks to the Author):

In the revised version now submitted to Nature Communications, the authors have addressed my technical concerns in a constructive manner. I have only suggestions for minor changes below, mainly about sharing the explanations with the reader.

The work's impact remains more ambivalent. The authors argue that they combine squeezing-enhanced measurement, feedback and opto-mechanical systems for the first time. This is correct, and as I wrote before, I understand this is a proof-of-principle demonstration. What makes the case less compelling, in my opinion, is that this demonstration is made so far away from where it could begin to matter, namely when coherent states come to their limits for whatever reason. The authors themselves have demonstrated 3x better performance (in terms of occupation) using coherent states in the same experiment, and others in the field some five orders of magnitude!

Non-classically enhanced sensing and feedback techniques should get closer if they ought to be of interest, especially if they are conceptually well-established. And they do: A good example is ref [8] cited by the authors: in this 2013 Nature Photonics paper, the LIGO Scientific collaboration claims that "With the injection of squeezed states, this LIGO detector demonstrated the best broadband sensitivity to gravitational waves ever achieved ..."

Therefore, while I certainly see the interest of using squeezed input states in opto- and electro-mechanical settings, I am personally not fully convinced of a broad impact of this work.

Further to comment #2:

To clarify the matter discussed in section C of the SI, it would be helpful to add to the SI the information that S_{XX} is the amplitude quadrature's power spectral density when the symbol is first introduced (presently it is just referred to as the "noise power spectral density of the incident field").

Further to comment #4:

The explanation of the authors about the nonlinear dependence of feedback gain on electronic gain is plausible, and important to know. I suggest to share this explanation from their rebuttal letter (amplification of adjacent modes) with the reader. On p. 21 of the SI, where they say the effect is explained, I only see references.

If this explanation is given, the fact that the squeezed and coherent probe feedbacks have a different dependence on the electronic gain is also plausible. So I strongly recommend to leave both curves, and explain that this is due to different levels of excitation of adjacent modes.

Reviewer #3 (Remarks to the Author):

The authors experimentally demonstrate that by using a non-classical probe field (in their case, squeezed light), a mechanical resonator can be cooled more efficiently. They provide convincing

evidence for that, backing it up with theoretical calculations.

The effect quantitatively is not so big, about 10% in terms of temperature, and at this point, much lower temperatures have been achieved by standard cooling techniques. The observations were also to anticipate from the theoretical point of view. Despite all of this, I totally agree with the authors that this is a proof of principle demonstration. The authors present the first demonstration of the effect of a non-classical light probe and provide a perspective how further research on non-classical probes could lead to results better than the current state-of-the-art of the field.

Given that the manuscript is clearly written, the presented evidence looks convincing to me, the manuscript has been already to the referees, and the response of the authors looks convincing, I recommend publication as is.

Final revisions for manuscript NCOMMS-16-18620-T

*Quantum enhanced feedback cooling of a mechanical oscillator using
nonclassical light*

Clemens Schäfermeier, Hugo Kerdoncuff, Ulrich B. Hoff, Hao Fu, Alexander Huck, Jan Bilek, Glen I. Harris, Warwick P. Bowen, Tobias Gehring, & Ulrik L. Andersen.

Kongens Lyngby, 7th October 2016

Dear Dr. Bishwanath Gaire,
thank you for the positive reply and suggestions to improve the manuscripts.
Reviewer 3 expressed her/his general agreement with the article, such that we reply to the comments of reviewer 2.

Reviewer 2, comment 2:

To clarify the matter discussed in section C of the SI, it would be helpful to add to the SI the information that S_{XX} is the amplitude quadrature's power spectral density when the symbol is first introduced (presently it is just referred to as the "noise power spectral density of the incident field").

Reply: The symbol S_{XX} is now introduced accordingly.

Reviewer 2, comment 4:

The explanation of the authors about the nonlinear dependence of feedback gain on electronic gain is plausible, and important to know. I suggest to share this explanation from their rebuttal letter (amplification of adjacent modes) with the reader. On p. 21 of the SI, where they say the effect is explained, I only see references.

If this explanation is given, the fact that the squeezed and coherent probe feedbacks have a different dependence on the electronic gain is also plausible. So I strongly recommend to leave both curves, and explain that this is due to different levels of excitation of adjacent modes.

Reply: Both curves are now included in the SI and the explanation given in our first response is contained in the same.

On behalf of the authors,
Clemens Schäfermeier